# Cluster Donation: How Future Healthcare Professionals Bound Certain Types of Tissues and Biomedical Research and How It Affects Their Willingness to Donate

**DOI:** 10.3390/healthcare11192636

**Published:** 2023-09-27

**Authors:** Jan Domaradzki, Marcin Piotr Walkowiak, Dariusz Walkowiak

**Affiliations:** 1Department of Social Sciences and Humanities, Poznan University of Medical Sciences, 60-806 Poznan, Poland; 2Department of Preventive Medicine, Poznan University of Medical Sciences, 60-781 Poznan, Poland; 3Department of Organization and Management in Health Care, Poznan University of Medical Sciences, 60-356 Poznan, Poland

**Keywords:** attitudes, biomedical research, cluster donation, future healthcare professionals, medical and healthcare students, tissue donation

## Abstract

Although biomedical research requires cooperation with a large number of donors, its success also depends on the input of healthcare professionals who play a crucial role in promoting biomedical research and influencing an individual’s decision to donate one’s biospecimens that are left over after a medical procedure. This work was aimed at investigating the correlation between medical and healthcare students’ willingness to donate a biospecimen, the type of tissues to be donated, and the type of biomedical research to be conducted. A population survey among medical and healthcare students enrolled at the Poznan University of Medical Sciences was conducted on their attitudes toward the donation of human biological material for research purposes. A total of 1500 students responded and completed the survey. The questionnaire used multiple-choice closed-ended questions designed to explore medical and healthcare students’ preferences for sharing particular types of tissues and donating to a particular type of biomedical research. It asked three questions: 1. Which type of tissue would people be willing to donate for research purposes? 2. Which organs would they be willing to donate after death? 3. What type of research would they be willing to donate to? While future healthcare professionals’ beliefs regarding certain types of tissues and research can influence their willingness to donate for research purposes, many students tend to think about the body and biomedical research in terms of clusters. Consequently, their willingness to donate certain tissues for a particular type of research can affect their decision to donate other tissues. Our data suggest that cluster thinking with regard to donation can be a predictor of people’s readiness to participate in the collection and management of biospecimens.

## 1. Introduction

In recent years, biomedical research has revolutionized science and provided the foundation for the progress of personalized medicine [1,2]. However, the success of biomedical research requires broad public cooperation and the creation of a substantial amount of high-quality biosample collections and associated clinical, genomic, and health information from a diverse range of people who would be willing to donate various types of tissues for research purposes [3,4,5,6,7]. Since most of these data are generated in academic research projects, clinical settings, genetic testing laboratories, or biobanks, biomedical research also requires the input of healthcare professionals (HCPs), including physicians, surgeons, pathologists, and nurses who can play a crucial role in obtaining patients’ consent to the collection of their biological material, the promotion of biomedical research, and building trust towards research institutions. HCPs are also in a key position to reduce donors’ ethical and moral concerns related to tissue donation and biomedical research [8,9,10,11].

Nevertheless, studies show that there is a discrepancy between people’s attitudes towards the donation of organs for transplantation and the actual donation of biospecimens to biomedical research. For example, although 93% of Polish citizens support the idea of organ transplant and 80% of adult Poles declare a willingness to donate their body organs to transplantation after death [12], few people are actually willing to donate their human biological material (HBM) to scientific research [13]. Moreover, even though many Poles are willing to donate their biospecimens for research purposes, their knowledge of biomedical research is low, and they often confuse participation in biomedical research with medical examinations [6,13,14,15].

For this reason, the role of HCPs in promoting biomedical research cannot be overestimated. Indeed, because HCPs are situated at the interface between biomedical research and possible donors, they can serve as recruiters and agents for biomedical research programs [9]. Moreover, although information on biomedical research is available through various channels, including the radio, television, magazines, social media, and the Internet, it is HCPs who are highly trusted by patients as reliable sources of information. Additionally, because HCPs can help donors resolve many ethical and moral dilemmas related to donation and biomedical research, they can reduce people’s concerns while also establishing trusting relationships with their patients [16,17]. Finally, HCPs can help their patients interpret genomic results and their implications for the patient-donor’s health and can thus influence patients’ decisions to donate their biospecimens for research purposes.

However, while the biomedical research knowledge of the nonmedical population is low, some research also suggests that members of the medical community do not possess adequate knowledge on the topic or may somehow feel uncomfortable in sending their patients for entry into biomedical research [8,18,19,20]. Unsurprisingly, research conducted among medical students also reveals knowledge deficits regarding biomedical research [21,22,23,24]. Thus, while future HCPs can promote biomedical research, their lack of knowledge or support for the data collection process can be a limiting factor. Meanwhile, since biomedical research is becoming increasingly important for precision medicine, it requires cooperation from competent healthcare professionals who can help to disseminate the idea of biomedical research among patients and their families and will support donations for research purposes. For that reason, it is important to assess medical and healthcare students’ attitudes towards participation in biomedical research. At the same time, while the system of donation is very complex and consists of many dimensions, including biological, cognitive, social, and institutional, which together influence the reasons why people decide to donate for research purposes [25], this work was aimed at investigating the correlation between medical and healthcare students’ willingness to donate a biospecimen and the type of tissues to be donated and the type of biomedical research to be conducted.

## 2. Material and Methods

### 2.1. Study Design

This research was designed as a self-administered, online population survey regarding medical and healthcare students’ willingness to donate various types of biospecimens for research purposes. Respondents were asked to complete a computer-assisted questionnaire using electronic devices.

### 2.2. Participants and Setting

The data presented here come from survey-based research carried out between December 2021 and February 2022 at the Poznan University of Medical Sciences (PUMS) in Poland and looks at medical and healthcare students’ attitudes toward the donation of human biological material (HBM) for research purposes. Participants were included if they were enrolled in PUMS and were eager to participate in the study. Invitation to participate in the study was posted on an online platform. Overall, 1500 students responded and completed the survey.

### 2.3. Research Tools

The questionnaire was elaborated upon according to the guidelines of the European Statistical System [26], and its validity was verified earlier in a pilot study. This research included data regarding medical and healthcare students’ willingness to donate certain types of tissues to various biomedical research. Apart from standard questions concerning students’ demographic characteristics, including sex, year of study, and faculty, the questionnaire used closed-ended, multiple-choice questions designed to explore medical and healthcare students’ preferences for sharing particular types of tissues and donating to a particular type of biomedical research. It explored three main questions: 1. Which of the following tissues would you donate for research purposes? 2. Which of the following tissues would you donate for research purposes after death? 3. What type of research would you be willing to donate to?

### 2.4. Data Collection

The structured questionnaire was posted on an online platform and distributed to all students of all faculties at PUMS (medicine, medical sciences, pharmacy, and health sciences) via a communication platform used at PUMS for educational purposes during the COVID-19 pandemic. Since this study was designed as a population survey, all students enrolled in PUMS received an invitation email and were informed about the study’s purpose, as well as the voluntary, anonymous, and confidential character of the study. Out of all the 5830 students approached, a total of 1500 (25.72%) students responded and completed the survey. At the same time, the social or demographic features of our sample were similar to those in the general population of students at PUMS.

All participants completed self-administered, computer-assisted questionnaires using electronic devices. Questionnaires took approximately 20 min to complete and were collected anonymously.

### 2.5. Ethical Issues

This study was performed in line with the principles of the Declaration of Helsinki [27]. Although according to local legislation and national guidelines on research involving human subjects, ethical approval, and research governance approval were not required, they were obtained from the PUMS Bioethics Committee (KB–926/21). The final version of the questionnaire was also approved by the University Student Council Board (USCB). Students received an invitation letter and were informed about the study’s purpose, as well as the voluntary, anonymous, and confidential character of the study. Additionally, all participants were informed that they could quit the survey at any given time and refuse to reveal information regarding their personal circumstances. All students who agreed to participate in the study provided their voluntary and informed consent.

### 2.6. Data Analysis

The data collected in the questionnaires were verified and checked for completeness, quality, and consistency and exported into the statistical package JASP (Version 0.18). Pearson’s Chi-square test was used to assess differences in the distribution of answers among socio-demographic groups. The results were presented as descriptive statistics. We developed a method for reducing the complexity of answers on individual thematic groups using principal component analysis (PCA) with an oblique (promax) rotation. PCA simplifies the complexity of high-dimensional data while retaining trends and patterns. We aimed to reduce the high-dimensional complexity of the respondents’ answers to just a few principal components that explain the majority of variation in those repertoires.

We tested our hypothesis that our respondents answering questions about tissue donations in the majority of their decisions can be assigned to a maximum of several groups. Reliability was assessed and compared for the three items using Cronbach’s α as a measure of internal consistency. We used a standard criterion of eigenvalues higher than 1 to determine the number of factors to retain for subsequent analyses. We interpreted the factors based on the variables with the highest factor loadings, focusing on those with factor loadings higher than 0.4 only. For the first item, Cronbach’s α was 0.704; for the second item, it was 0.830; and for the third item, it was 0.821. Kendall’s tau B was used to measure the correspondence between the respondents’ answers in each item.

Cluster analysis of tissue donations attitude was performed in Python 3.10 using Kmeans from the scikit-learn library version 0.0.post1. Subsequently, to characterize the formed clusters, each cluster was compared to the remaining clusters with respect to all variables, including demographic variables, to assess whether they exhibited differing distributions. For binary variables, Pearson’s Chi-square test was employed, while for Likert scale questions, the Kruskal–Wallis H-test, both tests were from SciPy 1.7.3. A statistical significance threshold of *p* < 0.05 was applied. The Sankey diagram was generated using plotly version 5.14.1.

## 3. Results

A total of 1500 completed and submitted questionnaires were analyzed (Table 1). The sample consisted of 1190 women (79.3%) and 310 men (20.7%), all of Polish origin. Although students represented all degree courses and years of study, over half were in their first or second year of study (29% and 22.8%, respectively), with the predominant majority of students coming from the medicine (24.1%), physiotherapy (13.8%), and pharmacy (12.4%) departments.

The PCA of respondents’ declarations to donate various types of tissues for research purposes revealed a two-factor solution (Figure 1A). The figure below presents Kendall’s tau B heatmap between answers (Figure 1B). Thus, the PCA suggests that there are two different ways to interpret tissue donation among respondents. Students who declared a willingness to donate reproductive tissues were more eager to donate embryonic cells left over after IVF procedures (rho = 0.575, *p* < 0.001), as well as all other types of tissue (rho = 0.291, *p* < 0.001). On the other hand, donating saliva, blood, and hair seemed to be treated on par with one another: respondents who declared a willingness to donate saliva also declared a willingness to donate blood (rho = 0.435, *p* < 0.001) and hair (rho = 0.353, *p* < 0.001) more often. However, they tended to refuse to donate any type of tissue (rho = −0.3, *p* < 0.001) more often than those who declared a willingness to donate other types of tissue. Similar results were obtained in the case of blood (rho = −0.333, *p* < 0.001) and hair (rho = −0.099, *p* < 0.001).

The PCA also showed that the way in which respondents declared which of the tissues they would be willing to donate for research purposes after death also reveals a two-factor solution (Figure 2A). Even though we are again dealing with two groups, the results of this figure differ from those of the previous one because the division into groups is different this time. The first group consists of those with extreme views: they are either ready to donate their entire body after death or refuse to hand over any part of it. The second group consists of those who accept the donation of only some selected tissues. It appears that some body parts go into certain clusters. For instance, the kidneys, lungs, and liver seem to be treated similarly as potential objects for donation, as indicated below (Figure 2B), which presents Kendall’s tau B heatmap between answers. The highest correlation was observed between the acceptance of liver and kidney donation (rho = 0.923, *p* < 0.001).

Finally, the PCA demonstrated that future HCPs’ declarations on the type of research they would be willing to donate their samples to can be divided accordingly and reveal a two-factor solution (Figure 3A). Once more, a two-way division emerges between those ready to donate their tissues to research on cancer pathogenesis, curable somatic diseases, incurable genetic diseases, and psychiatric disorders and those ready to donate them to research on intelligence, aggression and violence, reproductive cloning, as well as to commercial research. The figure below presents Kendall’s tau B heatmap between answers (Figure 3B). The highest correlation was observed between a willingness to donate tissues to research on intelligence and aggression and violence (rho = 0.754, *p* < 0.001).

The conducted analysis of the responses broken down into groups with specific socio-demographics (sex, faculty, year of study) showed a differentiation in this respect (Table 2). Many questions about biomedical research were answered differently by women than by men. Differences in replies also existed among students of different majors or years of study.

In Figure 4, we present a cluster analysis of students’ attitudes towards donation in relation to various types of biospecimens and different types of biomedical research. Additional socio-demographic characteristics in square brackets reflect correlates that became statistically significant, even though the algorithm did not consider them during the clustering process. The characterization of clusters, unless specified otherwise, is based on variables that made a particular cluster statistically different from the rest, not necessarily in terms of an absolute majority. This means that in a particular cluster, men were overrepresented in a field otherwise dominated by women or that in this generally supportive demographic towards research, researching traits like IQ or aggression was met with ambivalent feelings. The primary split (K = 2) was between those who were highly supportive of research and tissue donation and those who were skeptical about the whole concept. The enthusiastic group had a higher-than-typical representation of men or individuals studying medicine or biotechnology, while it was opposed by a group with an increased representation of women and physiotherapy students.

The division into three clusters led to the formation of an intermediary cluster, albeit lacking clear demographic characteristics. This intermediary cluster exhibited a generally moderate level of support, except in the context of cloning or commercial research. Further division (K = 4 and K = 5) did not result in any new arrangements but instead led to the initial splitting of the most research-accepting cluster and, subsequently, the splitting of the intermediary cluster. This suggests that the division at K = 3 reflected a genuine underlying pattern. Nonetheless, the division at K = 5 is presented as well, although the underlying differences in the split clusters were subtle, as the dividing features were only statistically significant in relation to their counterparts and not to the remaining clusters combined. The most research-accepting cluster was divided based on the willingness to donate tissues after death. The intermediary cluster underwent division based on the inclination to donate a specific tissue or the entire body. However, those willing to donate tissue, still in relation to the remaining clusters combined, exhibited clear support for less contentious research.

## 4. Discussion

Although people’s willingness to donate tissue for the purposes of research is mostly driven by their desire to support biomedical research [6], such a decision also depends on the donors’ knowledge about the donation system and biomedical research [21,28], declared system of values [29,30], religious beliefs [17,31], and various personality traits [14,32] and demographic characteristics of potential donors, including age, sex, education, and family and ethnic background [4,6,30,33,34]. However, previous experiences with the healthcare system [35], social trust in physicians and scientists [15], and perceived benefits and risks associated with biomedical research [36,37] also affect people’s attitudes toward their willingness to donate.

This should not be surprising because the system of donation involves a highly complex set of processes and interactions between various dimensions, which together influence the reasons why people make decisions to donate for research purposes [25]. Simultaneously, while there is a high level of acceptance for biomedical research and a willingness to donate to HBM for research purposes, there are some important barriers that can hinder peoples’ willingness to participate in donation. For example, there are many anxieties surrounding ethical, legal, and social issues (ELSI) related to the acquisition, storage, and sharing of biosamples donated for biomedical research, especially in terms of data protection, privacy, and confidentiality, as well as the commercialization of research results [38,39,40,41]. Additionally, there are many social, cultural, and psychological barriers that can hinder donation to biomedical research, including physical distance and the necessity to travel, registration requirements, moral and ethical concerns related to biomedical research, and contradictory information about the donation process [6].

Thus, drawing from Laura L. Machin et al.’s ‘pyramid of donation’ [25], which situates various body parts and body products within the hierarchy of systems involved in the donation process (tissues and cells, organs, biological systems, the person, family, community, culture, society, and nation) (Figure 5), this paper investigated how medical and healthcare students’ willingness to donate for research purposes interacts with certain body parts and particular types of biomedical research.

Thus, the results from this study are in line with previous findings that have shown that some of the donors’ concerns over donation are related to certain types of HBM and the purposes for which they are used. For example, Lewis et al. [4] demonstrated that while willingness to donate among members of the UK public was more influenced by the tissue type than the amount of tissue, the most controversial types of HBM include brain post mortem (29% of respondents would donate), eyes post mortem (35%), embryos (44%), spare eggs (48%), and sperm (58%). On the other hand, respondents were willing to donate their residual blood (92%), cancerous tissue (90%), fat (89%), skin tissue (88%), bone or cartilage (84%), and liver tissue (84%) left over after a medical procedure. Goodson and Vernon [3] demonstrated that healthy adult volunteers from a National Health Service dental practice in Newcastle were most likely to donate tissues obtained from the head and neck (74%) as well as ovarian or testicular tissues (71%), while the least popular were tissues of bone (50%), eye (54%), brain (58%), and lungs (58%). Similarly, a recent study conducted among the adult population in Poland demonstrated that people were more willing to donate urine (73.9%), blood (69.7%), hair, and tears (69.6%) but were reluctant to donate post-mortem brain fragments (20%), sperm (males; 36.4%), and egg cells (females; 39.6%) [15]. Finally, while senior healthcare students in Saudi Arabia declared a willingness to donate their blood (82%), saliva/sputum (77%), urine (70%), buccal swabs (66%), hair (67%), and toenails (49%), they were less likely to donate their own excess surgical tissue (43%) [23].

Similarly, this research confirms that the type of organ also influences people’s decision to donate after death. For example, many British respondents were reluctant to donate their brain (71% would not donate) and eyes (65%) but were more likely to donate their heart (73% would donate), liver (86%), lungs (86%), and kidneys (87%) [4]. Similarly, only 25% of healthcare students in Saudi Arabia would agree to donate a deceased family member’s organs or tissues [23].

Regarding the use of samples, this study is also in line with previous research conducted among Mexican-Americans in Texas [30] and adult Australians [42] that suggest that the most controversial research includes embryonic stem cell research, germ-line gene therapy, human cloning, the creation of immortalized cell lines, or human–animal hybrids. For example, Goodson and Vernon showed that while 82% of British respondents would consent to their tissues being used for cancer research, they were reluctant to donate their tissues for research on genetic disorders (65% would give consent), testing medicines (59%), and genetic cloning (26%) [3]. Other studies demonstrated that while people were willing to donate their tissues to research that would help science understand how our body fights disease (85% would donate) and how genetics influence disease susceptibility (87%), they were concerned that their samples could be used in combination with animals (34% would donate), for research conducted outside of the UK (35%), and in research involving cells from embryos (41%) [4]. Finally, some studies showed that respondents object to donating to research with a stigmatizing potential, i.e., on mental disorders, intelligence, or homosexuality [43], to research with commercial potential [44], and to research conducted abroad [45].

Simultaneously, this study shows that future HCPs enrolled in this study were prone to what we call cluster donation, i.e., while they declared a willingness to donate certain body parts but not others and were ready to provide their tissues or body parts for some research but not for other, they were thinking about donation to biomedical research in a collective way and thus grouped various types of tissues or research into clusters [46]. Their consent to donate certain types of HBM for certain types of research can affect their decision to donate or not donate tissues to other types of research that can be loosely related. Thus, this study shows that such cluster thinking about donation for research purposes can either be a stimulating or a limiting factor. The reason for this is that such cluster thinking is perhaps not so much related to logical arguments but is more driven by emotional associations, different cultural meanings attached to various body parts, and different risks or costs associated with various biomedical research.

This should come as no surprise because social and cultural beliefs about what should or should not be done to a body and its parts post mortem affect people’s attitudes toward tissue or organ donation and biomedical research. For example, while according to some cultural beliefs, a person is declared dead only when a particular organ no longer functions (heart or brain, or both), certain types of tissues or organs are perceived as the very essence of humanity (brain, heart, blood, eyes), while others are more defined in terms of function (hands, legs, breasts), and still others are valued for their functional or aesthetic significance (face, skin, hands) [15]. Indeed, while the brain is often defined as the most powerful organ in the human body and the essence of human existence, some people worry that it will not be treated with respect during an autopsy or insist that it should not be separated from the body, which should remain whole at the time of burial [47,48]. Likewise, because the heart is often seen as a repository of the soul, some people are reluctant to donate it after death. Other research suggests that, for some individuals, because the eyes represent the windows to the soul and are perceived as the most meaningful feature of a person, they are the least donated tissue [49,50]. Finally, people’s belief that blood is sacred or that there is a transfer of the spirit from the donor to the recipient can result in their negative attitude towards organ donation [51,52].

In parallel, this study demonstrates that factors such as gender, faculty, and academic year exerted discernible impacts on students’ proclivity towards tissue donation. Notably, statistically significant distinctions were detected concerning the specific tissue types and research domains for which students were inclined to offer their donations. Across the majority of research categories and various tissue types, male students and those pursuing medical and biotechnology disciplines exhibited the highest levels of willingness to contribute, whereas their female counterparts and students specializing in physiotherapy displayed comparatively lower inclinations toward such contributions.

However, Goodson and Vernon also demonstrated that people’s willingness to donate is influenced by their sex, as females were more willing to donate most types of body tissues for clinical research (including eye, head, and neck tissue and heart, brain, lung, or bone), but men were less restricted about the type of research their body tissues or organs would be used for (i.e., general knowledge, genetic disorders, testing medicines) [3]. Similarly, Lewis et al. [4] showed that the willingness to donate spare eggs and spare embryos was much more common among men who were under 55 years, from a higher socioeconomic group, White, not at all or moderately religious, and had tissue removed during a medical procedure. On the other hand, women who supported the donation of reproductive tissues were from a higher socioeconomic group, White, and had no religious affiliation. While preferences in the Polish population did not differ statistically for sex, age, education, religiousness, or trust in other people, it was engagement in charity, financial situation, and trust in physicians and scientists that influenced people’s willingness to donate [15]. Other research suggests that education level is also associated with higher awareness of bioethics, which affects people’s willingness to donate excess surgical tissue [23].

Boise et al. found out that among diverse ethnic groups, older age and Latino ethnicity were positive predictors for the willingness to become a brain donor, while African/African-American race was a negative predictor [47]. Another study by Hussen et al. demonstrated that the willingness to donate eyes among Ethiopians was associated with religion, educational level, and awareness [49]. High school education was also found to be a positive predictor for eye donation in India, while religious reasons and cultural beliefs were the main reasons for the lack of willingness to donate eyes [50]. Finally, older Chinese women living in Chicago’s Chinatown were more eager to donate hair or nails than blood [52].

Although, to the best of our knowledge, this is one of the few studies on the views and attitudes of future Polish healthcare practitioners on tissue donation for research purposes, it has a few limitations. Firstly, as this study was conducted during the COVID-19 pandemic, it impeded the recruitment process of participants. Consequently, the number of responses was rather small, and the results cannot be generalized to the entire population of medical and healthcare students in Poznan. Secondly, this study has a local dimension as only responses from medical and healthcare students from one Polish medical university were analyzed. Consequently, it would be desirable to compare these findings to findings from other locations in the country. Thirdly, because no specific questions regarding the meaning students attached to various types of tissues or research were asked, further in-depth studies would be required. Finally, as we analyzed only hypothetical, and not actual, decisions to donate, it should be stressed that actual behaviors and intentions often differ.

However, despite these limitations, some advantages of this study should also be acknowledged. Most importantly, as there is a scarcity of previous work on this topic, this research fills the gap in the literature regarding the views of future HCPs on tissue donation for research purposes. Moreover, providing the concept of “cluster donation” brings new insights for medical and healthcare students’ personal beliefs on various types of tissues and/or research that can influence their role in the promotion of biomedical research.

## 5. Conclusions

While there are some psychosocial factors that can affect people’s attitudes toward donation for research purposes, many of these issues can be addressed by HCPs and university academics through effective communication and patient education [9,10,11]. As knowledgeable and reliable sources of information and guidance on biomedical research, HCPs are the best advisers for issues concerning biotechnology and can play a pivotal role in promoting biomedical research and positively influencing an individual’s decision to donate HMB left over after a medical procedure for research purposes [53]. However, a lack of HCPs’ support for biomedical research can hinder the process of collecting and processing biospecimens [8,9].

Indeed, this study has shown that medical and healthcare students’ beliefs regarding certain types of tissues or organs and research can influence their willingness to donate for research purposes. Moreover, because many future healthcare professionals tend to think about the body and biomedical research in terms of clusters, it can be a predictor of their readiness and willingness to support biomedical research and participation in the collection and management of biospecimens. Such beliefs can also affect students’ readiness to approach patients or their families, who could share their biospecimens left over after a medical procedure and hinder their ability to recruit tissue and organ donors.

Thus, apart from public education campaigns that address the role and function of biomedical research, there is still an urgent need to raise medical and healthcare students’ awareness about their important role in research activities. This is of key importance because, as future healthcare workers, they will be responsible not only for caring for patients but also for the promotion of biomedical research. Put together, to overcome medical and healthcare students’ ambivalent attitudes toward tissue donation for research purposes, we recommend that they should receive more education and awareness about the importance of furthering biomedical research and precision medicine. To achieve that, the following guidelines should be implemented:In order to raise medical and healthcare students’ awareness and interest in biomedical research and donation, university curricula in Poland should include teaching programs on such issues;Because biomedical research and donation for research purposes provoke many ethical and moral dilemmas among the donors, future healthcare professionals should also be familiarized with the ethical and legal framework for donation, (non-verbal) communication and active listening skills, the role of cultural belief systems on tissue and organ donation, and education of the general public on donation;Because many healthcare professionals feel reluctant to approach their patients or their families who could possibly share their HBM for research purposes, medical and healthcare students should be trained on how to effectively recruit tissue and organ donors;Additionally, e-learning modules, counting tutorial lecturer videos, webinars, online postgraduate specialization courses, healthcare professional awareness, and training programs on biomedical research and donation should also be organized.

## Figures and Tables

**Figure 1 healthcare-11-02636-f001:**
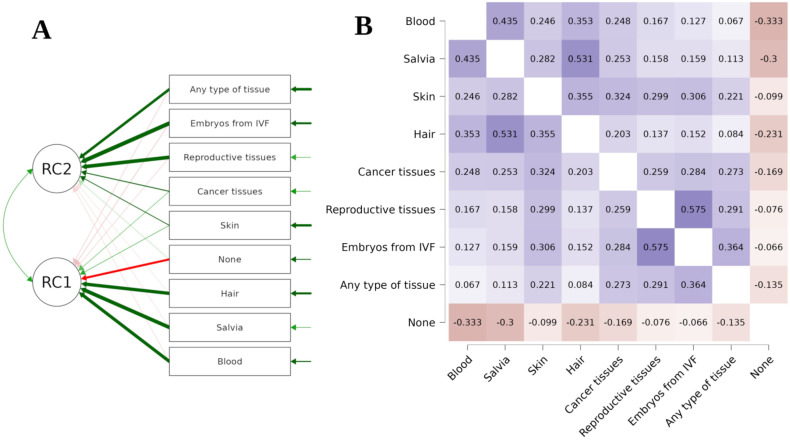
Students’ willingness to donate in relation to the type of tissue.

**Figure 2 healthcare-11-02636-f002:**
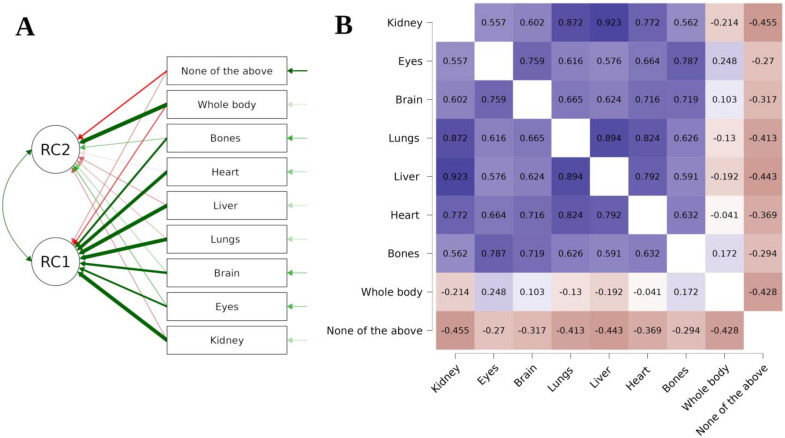
Students’ willingness to donate after death.

**Figure 3 healthcare-11-02636-f003:**
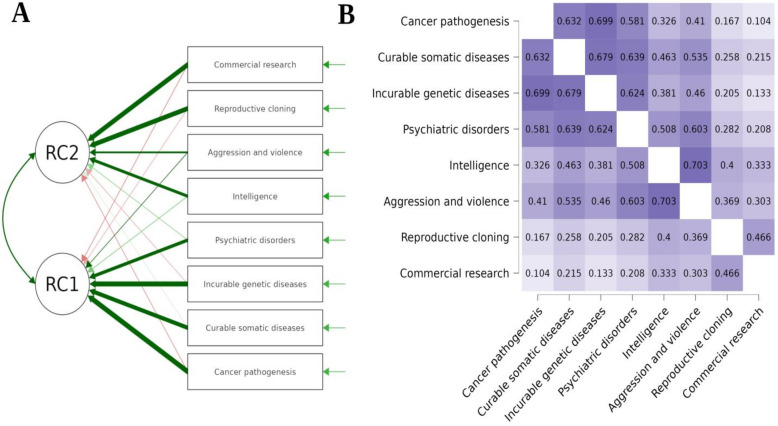
Students’ willingness to donate in relation to the type of research.

**Figure 4 healthcare-11-02636-f004:**
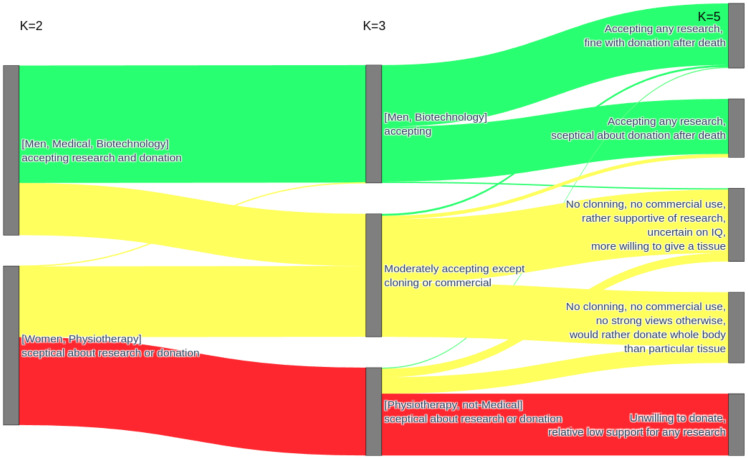
Cluster analysis of students’ attitudes towards donation in relation to different types of tissue and research.

**Figure 5 healthcare-11-02636-f005:**
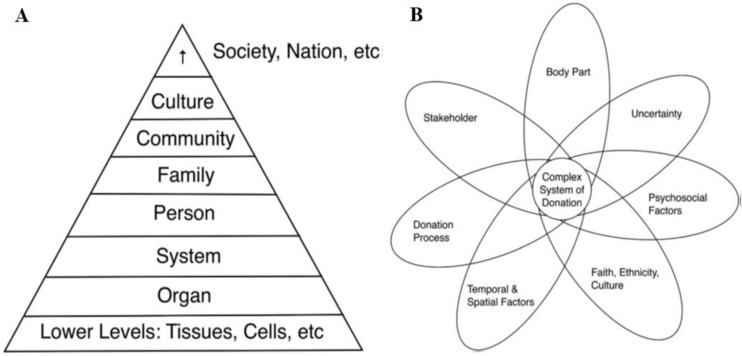
Machin et al.’s pyramid of donation (**A**) and their model of complexity of the donation system (**B**). Machin et al.’s ‘pyramid of donation’ situates various body parts and body products within the hierarchy of systems involved in the donation process (tissues and cells, organs, biological systems, the person, family, community, culture, society, and nation). Simultaneously, they distinguish seven dimensions that affect the system of donation: body part, donation process and uncertainty, faith, ethnicity and culture, psychosocial factors, stakeholder, and temporal and spatial factors [25].

**Table 1 healthcare-11-02636-t001:** Socio-demographic characteristics of respondents.

Characteristics	N (%)
*Sex*	
Female	1190 (79.3)
Male	310 (20.7)
*Year of study*	
1	435 (29)
2	341 (22.8)
3	240 (16)
4	218 (14.5)
5	221 (14.7)
6	45 (3)
*Faculty*	
Medicine	362 (24.1)
Physiotherapy	207 (13.8)
Nursing	143 (9.5)
Pharmacy	186 (12.4)
Electroradiology	61 (4.1)
Medical analytics	70 (4.7)
Dentistry	55 (3.7)
Midwifery	95 (6.3)
Medical rescue	26 (1.7)
Public health	53 (3.5)
Dietetics	49 (3.3)
Medical biotechnology	47 (3.1)
Other	146 (9.8)

**Table 2 healthcare-11-02636-t002:** The impact of socio-demographic characteristics on students’ willingness to donate.

	Sex	Faculty	Year of the Study
*Which of the following organs would you donate for research purposes?*			
Blood	ns	ns	ns
Salvia	ns	ns	ns
Skin	<0.001	<0.01	ns
Hair	ns	ns	ns
Cancer tissues	ns	<0.01	<0.001
Reproductive tissues (sperm, eggs)	<0.001	<0.05	ns
Embryonic cells left over after an IVF procedure	ns	<0.01	ns
Any type of tissue that is left over after a medical procedure	<0.05	<0.001	ns
None of the above	ns	<0.05	ns
*Which of the following organs would you donate for research purposes after death?*			
Kidney	ns	ns	ns
Eyes	ns	ns	ns
Brain	ns	ns	ns
Lungs	ns	ns	ns
Liver	ns	ns	ns
Heart	ns	ns	ns
Bones	ns	ns	ns
Whole body	ns	<0.01	<0.01
None of the above	ns	ns	ns
*To what type of research would you be willing to donate?*			
Research on the pathogenesis of cancer	<0.001	ns	ns
Research on curable somatic diseases	<0.001	<0.05	<0.05
Research on incurable genetic diseases	ns	ns	ns
Research on psychiatric disorders, i.e., schizophrenia, depression	<0.05	<0.001	ns
Research on intelligence	<0.001	<0.05	ns
Research on aggression and violence	<0.001	<0.05	ns
Research on reproductive cloning	<0.001	<0.05	ns
Commercial research	<0.05	ns	ns

ns: statistically not significant.

## Data Availability

Data generated as part of this study are available from the corresponding author on reasonable request.

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
