# Peer review of "Cluster Donation: How Future Healthcare Professionals Bound Certain Types of Tissues and Biomedical Research and How It Affects Their Willingness to Donate"

_healthcare, 2023, doi:10.3390/healthcare11192636_

Round 1

Reviewer 1 Report

This manuscript discusses healthcare medical students' willingness to donate their tissues or body parts for biomedical research. It raises several research questions, such as which tissues and body parts they are willing to donate and for what kind of research. The narrative appears intriguing. However, if the author could extend the discussion to include the general population rather than focusing solely on healthcare medical students (a relatively small and specific group), taking into consideration factors like education, gender, family background, and religion, it could make the story more compelling. Additionally, the credibility of the results could be improved by incorporating more analyses, such as clustering, robust analysis, and multivariate analysis.

Author Response

Dear Reviewer,

First of all, we would like to express our gratitude to you for giving us the opportunity to revise and resubmit our paper. We are also indebted to your valuable suggestions and helpful comments. We hope that this revised paper is more consistent owing to your willingness to help. We have been convinced by all of your arguments and we are very grateful to you for pointing these things out. We believe that we have also answered all of your questions. Still, should it happen that we have missed and/or misunderstood any vital comment, we would be more than happy to promptly address any edits and further revise our article.

Below, we detail the changes that we have made in accordance with your suggestions and comments (please note that in the revised manuscript we have taken the liberty of marking the most important changes in the colour red to facilitate their checking).

  1. Reviewer’s comment:

This manuscript discusses healthcare medical students’ willingness to donate their tissues or body parts for biomedical research. It raises several research questions, such as which tissues and body parts they are willing to donate and for what kind of research. The narrative appears intriguing. However, if the author could extend the discussion to include the general population rather than focusing solely on healthcare medical students (a relatively small and specific group), taking into consideration factors like education, gender, family background, and religion, it could make the story more compelling.

Authors’ response:

Thank you, we appreciate your comment. However, we wish to explain that since our work was aimed at investigating the correlation between medical and healthcare students’ willingness to donate and the type of tissues or body part to be donated and the type of biomedical research to be conducted, in the Discussion section we have focussed solely on research that asked questions regarding people’s attitudes toward donation certain types of HBM (including donation after death) and/or the various type of biomedical research for which they are used. Thus, we did not include the abundance of other research that did not ask such questions, but instead assessed either people’s knowledge about biomedical research, theirs motivations to share HBM for research purposes or attitudes towards ethical issues related to donation (i.e. preferred type of consent).

At the same time, we wish to explain that while in the Discussion section we do refer to one study on the willingness to donate biospecimens among senior healthcare students’ in Saudi Arabia [23], the vast majority of studies cited in this section were performed on general population, not on medical students. For example a stud by Lewis at al. [4] was conducted on members of the UK public. A study by Goodson and Vernon [3] describes attitudes of healthy adult volunteers from a National Health Service dental practice in Newcastle. A study by Majchrowska et al. [15] was carried out on a group of 1,100 people over 18 years of age, representing the adult population of Poland. Another two studies cited were conducted among Mexican-Americans in Texas [30] and members of Australian general public [40]. Finally, we also cite studies carried out on members of adult Jewish population [41], representatives of Chicago community [42] and Canadians [43].

However, having been persuaded by your concerns, we have referred to these populations in the Discussion section to make it more clear. This has been done on pages 10-11 (lines 315-343). Additionally, following your suggestion we have added a paragraph describing the impact of sociodemographic factors on people’s willingness to donate various types of tissues. This has been done on pages 11-12 (lines 384-406). Thus after revision it now says:

“In parallel, this study demonstrates that factors such as gender, faculty, and academic year exerted discernible impacts on students' proclivity towards tissue donation. Notably, statistically significant distinctions were detected concerning the specific tissue types and research domains for which students were inclined to offer their donations. Across the majority of research categories and various tissue types, male students and those pursuing medical and biotechnology disciplines exhibited the highest levels of willingness to contribute, whereas their female counterparts and students specializing in physiotherapy displayed comparatively lower inclinations toward such contributions

However, also Goodson and Vernon demonstrated that people’s willingness to donate is influenced by their sex, as females were more willing to donate most type of body tissues for clinical research (including eye, head and neck tissue, heart, brain, lung or bones), but men were less restricted about the type of research their body tissues or organs would be used for (i.e. general knowledge, genetic disorders, testing medicines) [3]. Similarly, Lewis et al. [4] showed that the willingness to donate spare eggs and spare embryos was much more common among men who were under 55 years, from a higher socioeconomic group, white, not at all or moderately religious, and had tissue removed during a medical procedure. On the other hand, women who supported donation of reproductive tissues were be from a higher socioeconomic group, white and had no religious affiliation. While preferences in Polish population did not differ statistically for sex, age, education, religiousness or trust in other people, it was engagement in charity, financial situation and trust in physicians and scientists that influenced people’s willingness to donate [15]. Other research suggest that also education level associated with higher awareness of bioethics affects people willingness to donate excess surgical tissue [23].”

We hope this explanation satisfies your concerns. At the same time, we are grateful to you for bringing our attention to this point.

  1. Reviewer’s comment:

Additionally, the credibility of the results could be improved by incorporating more analyses, such as clustering, robust analysis, and multivariate analysis.

Authors’ response:

Having been persuaded by your concerns we have provided cluster cluster analysis of students’ attitudes towards donation in relation to various types of biospecimen and different type of biomedical research. This has been done on pages 8-9 (lines 242-273). Thus, it now says:

“In Figure 4, we present a cluster analysis of students’ attitudes towards donation in relation to various types of biospecimen and different type of biomedical research. Additional socio-demographic characteristics in square brackets reflect correlates that became statistically significant, even though the algorithm did not consider them during the clustering process. The characterization of clusters, unless specified otherwise, is based on variables that made a particular cluster statistically different from the rest, not necessarily in terms of an absolute majority. This means that in a particular cluster, men were overrepresented in a field otherwise dominated by women, or that in this generally supportive demographic towards research, traits like researching IQ or aggression were met with ambivalent feelings. The primary split (K=2) was between those who were highly supportive of research and tissue donation and those who were skeptical about the whole concept. The enthusiastic group had a higher-than-typical representation of men or individuals studying medicine or biotechnology, while it was opposed by a group with an increased representation of women and physiotherapy students.

The division into three clusters led to the formation of an intermediary cluster, albeit lacking clear demographic characteristics. This intermediary cluster exhibited a generally moderate level of support, except in the context of cloning or commercial research. Further division (K=4 and K=5) did not result in any new arrangements but instead led to the initial splitting of the most research-accepting cluster and subsequently the splitting of the intermediary cluster. This suggests that the division at K=3 reflected a genuine underlying pattern. Nonetheless, the division at K=5 is presented as well, although the underlying differences in the split clusters were subtle, as the dividing features were only statistically significant in relation to their counterparts and not to the remaining clusters combined. The most research-accepting cluster was divided based on the willingness to donate tissues after death. The intermediary cluster underwent division based on the inclination to donate a specific tissue or the entire body. However, those willing to donate tissue, still in relation to the remaining clusters combined, exhibited clear support for less contentious research.”

[while the Figure 4 with cluster analysis of students’ attitudes towards donation in relation to different type of tissue and research could not be past here it can be found both in the file attached and the manuscript itself]

We hope this explanation satisfies your concerns. At the same time, we are grateful to you for bringing our attention to this point.

Additionally, in the Data Analysis section we have provide a short description of methods used in this analysis. This has been done on page 4 (lines 165-172). Thus, it now says:

“Cluster analysis of tissue donations attitude was performed in Python 3.10 using Kmeans from the scikit-learn library version 0.0.post1. Subsequently, to characterize the formed clusters, each cluster was compared to the remaining clusters with respect to all variables, including demographic variables, to assess whether they exhibited differing distributions. For binary variables, Pearson’s Chi-square test was employed, while for Likert scale questions, the Kruskal-Wallis H-test, both tests were from scipy 1.7.3. A statistical significance threshold of p<0.05 was applied. The Sankey diagram was generated using plotly version 5.14.1.”

We hope this explanation satisfies your concerns. At the same time, we are grateful to you for bringing our attention to this point.

All in all, again we wish to thank you for all your valuable suggestions and willingness to help us in improving our manuscript. At the same time, while we hope our revisions satisfy your concerns and that this revised paper is more consistent owing to your willingness to help, we would like to explain that although we have addressed all of your remarks, we have also tried to compress some responses in order to not exceed the word limit too much. This is especially since our manuscript was already rather long. Thus, while we have tried to fulfil your requirements and address all of your suggestions, we have tried to strike a happy balance between these two demands. However, if you find some of our responses insufficient, we will happily further revise our paper to make it even more reader-friendly.

Sincerely,

Authors

Reviewer 2 Report

Comments

This is an interesting subject as biospecimen donation is the crux of many biomedical research. The authors dealt with factors that may affect the donations of such biospecimen by donors. To make the work better, I have made few suggestions as shown below.

Abstract

The sampling method should be stated here. The method of asking the question should be stated here too.

Line 25- Use “with regards to” instead of “in regard to”

Introduction

Line 46- Towards

Line 50- Write HBM in full the first time you are using it here

Line 62- Put “,” after research

Line 72- Put “,” after research

Line 76- Remove “as to”

Line 76-78: This clause “this study aimed to assess whether future HCPs’ willingness to donate depends on the type of tissues and biomedical research conducted by scientific institutions.” Is not clear enough. Are HCPs the ones donating or the donors? The aim of the work should be written in a clear and simple form for easy comprehension by the reader.

Here is my suggestion

 “This work was aimed at investigating the correlation between the willingness to donate a biospecimen and the type of tissues to be donated/type of biomedical research to be conducted”

Materials and methods

Line 82- “a self-administered”. The research design is not clear. In your introduction, you talked about HCPs; but in the design, you talked about using medical students. Medical students are neither HCPs nor donors. They should be clarity on the population involved in the research. For now, it is confusing. What was the age bracket of the chosen population? Why choose the age bracket.? Why choose future HCPs

Line 85- How was the sample size of 1500 determined?

Line 105- use “Structured questionnaire” instead of just questionnaire since the questions are predetermined.  

Line 131- The results were presented as descriptive statistics

Results

Instead of using Right and Left for the figures, use “A and B” for the figures in each panel. For example, use Figure 1A, and 1B, not Figure 1, Left and Figure 1, Right.

Discussion

Line 338- showed or influenced? This is not clear

Line 343- firstly not first

Line 346- Secondly not “Second”

Line 349- Thirdly not third

Conclusion

Line 380- instead of “All in all”, use “Put together”.

In your conclusion, you used future HCPs as though they were already HCPs. You conclusion has to be entirely based on your findings not speculations or extrapolations.

General comments

You need overall minor grammatical corrections

The research designed has to be made clearer

-

Author Response

Dear Reviewer,

First of all, we would like to express our gratitude to you for giving us the opportunity to revise and resubmit our paper. We are also indebted to your valuable suggestions and helpful comments. We hope that this revised paper is more consistent owing to your willingness to help. We have been convinced by all of your arguments and we are very grateful to you for pointing these things out. We believe that we have also answered all of your questions. Still, should it happen that we have missed and/or misunderstood any vital comment, we would be more than happy to promptly address any edits and further revise our article.

Below, we detail the changes that we have made in accordance with your suggestions and comments (please note that in the revised manuscript we have taken the liberty of marking the most important changes in the colour green to facilitate their checking).

  1. Reviewer’s comment:

This is an interesting subject as biospecimen donation is the crux of many biomedical research. The authors dealt with factors that may affect the donations of such biospecimen by donors. To make the work better, I have made few suggestions as shown below.

Authors’ response:

Thank you, we appreciate your positive feedback. We are also thankful to your valuable and helpful comments and for giving us the opportunity to revise our paper.

  1. Reviewer’s comment:

Abstract

The sampling method should be stated here. The method of asking the question should be stated here too.

Authors’ response:

While we are indebted to you for this prospective comment we wish to explain that as the questionnaire was posted in an online communication platform used at PUMS an invitation letter was sent to all students enrolled at PUMS. However, since many students do not use their university email accounts not all students responded and completed the survey. Thus, while we were unable to assess the exact number of students who received an invitation, to our surprise, we have received exactly 1,500 questionnaires. At the same time, having been persuade by your concerns we have provided this explanation in the Data Collection section on page 3, lines 125-127. Thus after revision it now says (the revised part is marked in the colour green):

“2.4. Data Collection

The structured questionnaire was posted in an online platform and distributed to all students of all faculties at PUMS (medicine, medical sciences, pharmacy, and health sciences) via a communication platform used at PUMS for educational purposes during the COVID-19 pandemic. All students received an invitation email and were informed about the study’s purpose, as well as the voluntary, anonymous, and confidential character of the study. A total of 1,500 students responded and completed the survey.

All participants completed self-administered, computer-assisted questionnaires using electronic devices. Questionnaires took approximately 20 minutes to complete and were collected anonymously.”

Additionally, following your advice we have clarified the method of asking the question. Thus, we explain that the questionnaire used multiple choice closed-ended questions designed to explore medical students’ preferences for sharing particular type of tissues and donation to a particular type of biomedical research. This has been done on page 1, lines 23-25. Thus after revision it now says (the revised part is marked in the colour green):

Abstract: Although biomedical research requires cooperation with a large number of donors its success also depends on the input of healthcare professionals who play a crucial role in promoting biomedical research and influencing an individual’s decision in donating one’s biospecimens left over after a medical procedure. This work was aimed at investigating the correlation between medical and healthcare students’ willingness to donate a biospecimen and the type of tissues to be donated and the type of biomedical research to be conducted. We conducted a survey among 1,500 Polish medical and healthcare students on their attitudes toward the donation of human biological material for research purposes. The questionnaire used multiple choice closed-ended questions designed to explore medical and healthcare students’ preferences for sharing particular type of tissues and donation to a particular type of biomedical research. It asked three questions: 1. Which type of tissue would people be willing to donate for research purposes?, 2. Which organs would they be willing to donate after death?, and 3. What type of research would they be willing to donate to? While future healthcare professionals’ beliefs regarding certain types of tissues and research can influence their willingness to donate for research purposes many students tend to think about the body and biomedical research in terms of clusters. Consequently, their willingness to donate certain tissues for a particular type or research can affect their decision to donate other tissues. Our data suggest that cluster thinking with regards to donation can be a predictor of people’s readiness to participate in the collection and management of biospecimens.”

We hope this revision satisfies your concerns. At the same time, we are grateful for pointing this out.

  1. Reviewer’s comment:

Line 25- Use “with regards to” instead of “in regard to”

Authors’ response:

Following you suggestion we have revised this sentence accordingly. This has been done on page 1, line 33.

  1. Reviewer’s comment:

 Introduction

Line 46- Towards

Line 50- Write HBM in full the first time you are using it here

Line 62- Put “,” after research

Line 72- Put “,” after research

Line 76- Remove “as to”

Authors’ response:

Thank you, all these linguistic errors have been revised accordingly.

  1. Reviewer’s comment:

Line 76-78: This clause “this study aimed to assess whether future HCPs’ willingness to donate depends on the type of tissues and biomedical research conducted by scientific institutions.” Is not clear enough. Are HCPs the ones donating or the donors? The aim of the work should be written in a clear and simple form for easy comprehension by the reader.

Here is my suggestion

“This work was aimed at investigating the correlation between the willingness to donate a biospecimen and the type of tissues to be donated/type of biomedical research to be conducted”

Authors’ response:

Having been persuaded by your concerns we have modified the aim of our study. This has been done on page 2, lines 88-91. Thus after revision it now says (the revised part is marked in the colour green):

“At the same time, while the system of donation is very complex and consists of many dimensions, including biological, cognitive, social, and institutional, which together influence the reasons why people decide to donate for research purposes [25], this work was aimed at investigating the correlation between medical and healthcare students’ willingness to donate a biospecimen and the type of tissues to be donated and the type of biomedical research to be conducted.”

Additionally, it has been also revised in the Abstract (page 1, lines 18-21) We hope it is now more consistent and clearer owing your willingness to help. At the same time, we are grateful for pointing this out.

  1. Reviewer’s comment:

Materials and methods

Line 82- “a self-administered”. The research design is not clear.

Authors’ response:

While we appreciate this comment we wish to explain that a self-administered survey is a data-collection process where the researcher is entirely absent. Thus, our study was explicitly designed as an online survey to be completed by respondents without an interviewer’s assistance. However, in accordance with your suggestion we have reformulated the study design section to make it more clear. This has been done on pages 2-3, lines 95-98. Thus after revision it now say (the revised part is marked in the colour green):

“2.1. Study Design

This research was designed as a self-administered, online survey regarding medical and healthcare students’ willingness to donate a various type of biospecimens for research purposes. Respondents were asked to complete computer-assisted questionnaire using electronic devices.”

We hope this explanation satisfies your concerns. At the same time, we are grateful to you for bringing our attention to this point.

  1. Reviewer’s comment:

In your introduction, you talked about HCPs; but in the design, you talked about using medical students. Medical students are neither HCPs nor donors. They should be clarity on the population involved in the research. For now, it is confusing. What was the age bracket of the chosen population? Why choose the age bracket.? Why choose future HCPs

Authors’ response:

While we are indebted to you for this prospective comment, we wish to stress that both in the title and abstract we explicitly refer to “future healthcare professionals”. We also declare that in this study we aimed in assessing “how medical and healthcare students’ willingness to donate interacts with certain type of tissues or body parts, and biomedical research”. Thus, while indeed in the first part of the introduction we write about the role of healthcare professionals in promoting biomedical research (paragraphs 1-3), in the penultimate paragraph we refer strictly to medical and healthcare students and stress that previous research demonstrated that medical students’ knowledge on biomedical research is low and that such knowledge deficits can negatively affect their willingness to promote biomedical research. Thus, while we agree that medical and healthcare students are neither HCPs nor donors, we intended to contextualize our research by stressing how important it is to asses medical and healthcare students’ attitudes towards donation for research purposes. However, having been persuaded by your concerns we have clarified the reason for choosing future HCPs. Thus, on page 2 (lines 80-85) we explain that while the success of biomedical research requires not only large number of donors but also the input of healthcare professionals, it is important to asses medical and healthcare students’ attitudes towards participation in biomedical research, because in the future they will play a crucial role in promoting biomedical research and approaching their patients who could possibly share their HBM left over after a medical procedure for research purposes. Thus, after revision in now says:

“However, while the biomedical research knowledge of the nonmedical population is low, some research also suggests that members of the medical community do not possess adequate knowledge on the topic or may somehow feel uncomfortable in sending their patients for entry into biomedical research [8,18-20]. Unsurprisingly, research conducted among medical students also reveals knowledge deficits regarding biomedical research [21-24]. Thus, while future HCPs can promote biomedical research, their lack of knowledge or support for the data collection process can be a limiting factor. Meanwhile, since biomedical research are becoming increasingly important for the precision medicine they require cooperation from competent healthcare professionals who can help to disseminate the idea of biomedical research among patients and their families and will support donation for research purposes. For that reason, it is important to asses medical and healthcare students’ attitudes towards participation in biomedical research. At the same time, while the system of donation is very complex and consists of many dimensions, including biological, cognitive, social, and institutional, which together influence the reasons why people decide to donate for research purposes [25], this work was aimed at investigating the correlation between medical and healthcare students’ willingness to donate a biospecimen and the type of tissues to be donated and the type of biomedical research to be conducted.”

Finally, as for your remark regarding the age bracket we wish to explain that while there are no age limits to study in Poland, the vast majority of students are 19 years old when heading into their first year. The average graduating age is 24 (25 for medical students as the medicine course in Poland takes six years). For that reason, we have decide to focus on year of study rather than students’ age. We hope this explanation satisfies your concerns. At the same time, we are grateful to you for bringing our attention to this point.

  1. Reviewer’s comment:

Line 85- How was the sample size of 1500 determined?

Authors’ response:

As noted in point one of this cover letter the sample size was not predetermined but instead the questionnaire was posted in an online communication platform used at PUMS for educational purposes during the COVID-19 pandemic, and all students enrolled in PUMS received an invitation email to participate. However since many students do not use their university email accounts not all students responded and completed the survey. At the same time, to our surprise, we have received exactly 1,500 questionnaires. Thus, on page 3, lines 127-128 we explain that a total of 1,500 students responded and completed the survey:

“2.4. Data Collection

The structured questionnaire was posted in an online platform and distributed to all students of all faculties at PUMS (medicine, medical sciences, pharmacy, and health sciences) via a communication platform used at PUMS for educational purposes during the COVID-19 pandemic. All students enrolled in PUMS received an invitation email and were informed about the study’s purpose, as well as the voluntary, anonymous, and confidential character of the study. A total of 1,500 students responded and completed the survey.

All participants completed self-administered, computer-assisted questionnaires using electronic devices. Questionnaires took approximately 20 minutes to complete and were collected anonymously.”

We hope this explanation satisfies your concerns.

  1. Reviewer’s comments:

Line 105- use “Structured questionnaire” instead of just questionnaire since the questions are predetermined.  

Line 131- The results were presented as descriptive statistics

Authors’ response:

Thak you, it has been revised accordingly.

  1. Reviewer’s comments:

Results

Instead of using Right and Left for the figures, use “A and B” for the figures in each panel. For example, use Figure 1A, and 1B, not Figure 1, Left and Figure 1, Right.

Authors’ response:

Thak you, we have revised it accordingly.

  1. Reviewer’s comments:

Discussion

Line 338- showed or influenced? This is not clear

Line 343- firstly not first

Line 346- Secondly not “Second”

Line 349- Thirdly not third

Authors’ response:

Thak you, all these linguistic errors have been revised accordingly.

  1. Reviewer’s comments:

Conclusion

Line 380- instead of “All in all”, use “Put together”.

In your conclusion, you used future HCPs as though they were already HCPs. You conclusion has to be entirely based on your findings not speculations or extrapolations.

Authors’ response:

Having been persuaded by your concerns following your advice we have reformulated this part of our manuscript. This has been done on pages 12-13, lines 437-444. Thus, after revision it now says:

“While there are some psychosocial factors that can affect people’s attitudes toward donation for research purposes many of these issues can be addressed by HCPs and university academics through effective communication and patient education [9-11]. As knowledgeable and reliable sources of information and guidance on biomedical research, HCPs are the best advisers for issues concerning biotechnology and can play a pivotal role in promoting biomedical research and positively influencing an individual’s decision to donate HMB left over after a medical procedure for research purposes [51]. However, a lack of an HCPs’ support for biomedical research can hinder the process of collecting and processing biospecimens [8,9].

Indeed, this study has shown that future medical and healthcare students’ beliefs regarding certain types of tissues or organs and research can influence their willingness to donate for research purposes. Moreover, because many future healthcare professionals tend to think about the body and biomedical research in terms of clusters it can be a predictor of their readiness and willingness to support biomedical research and participation in the collection and management of biospecimens. Such beliefs can also affect students’ readiness to approach patients or their families, who could share their biospecimens left over after a medical procedure, and hinder their ability to recruit tissue and organ donors.”

We hope that this revised paper is more consistent owing to your willingness to help. At the same time, we are grateful for pointing this out.

Additionally, following another review’s advice we have formulated several suggestions that could improve students’ awareness and interest in biomedical research and donation for research purposes. This has been done on page 13, lines 452-468. Thus, after revision it now says:

“Thus, apart from public education campaigns that address the role and function of biomedical research there is still an urgent need to raise medical and healthcare students’ awareness about their important role in research activities. This is of key importance, because as future healthcare workers they will be responsible not only for caring over patients but also for the promotion of biomedical research. Put together, to overcome medical and healthcare students’ ambivalent attitudes toward tissues donation for research purposes we recommend that they should receive more education and awareness about the importance of furthering biomedical research and precision medicine. To achieve that the following guidelines should be implemented:

  1. In order to rise medical and healthcare students’ awareness and interest in biomedical research and donation, university curricula in Poland should include teaching programs on such issues.
  2. Because biomedical research and donation for research purposes provoke many ethical and moral dilemmas among the donors, future healthcare professionals should be also familiarized with ethical and legal framework for donation, (non-verbal) communication and active listening skills, the role of cultural belief systems on tissue and organ donation, and education of general public on donation.
  3. Because many healthcare professionals feel reluctant to approach their patients who could possibly share their HBM for research purposes, or their families, medical and healthcare students should be trained how to effectively recruit tissue and organ donors.
  4. Additionally, e-learning modules, counting tutorial lecturer videos, webinars, online postgraduate specialization courses, healthcare professional awareness and training programs on biomedical research and donation should be also organized.”

We hope this explanation satisfies your concerns. At the same time, we are grateful to you for bringing our attention to this point.

All in all, again we wish to thank you for all your valuable suggestions and willingness to help us in improving our manuscript. We hope that our revisions satisfy your concerns and that this revised paper is more consistent owing to your willingness to help. However, if we have missed and/or misunderstood any vital comment, we would be more than happy to further revise our article.

Reviewer 3 Report

The authors conducted an interesting study regarding the will of Polish healthcare students to participate in research activities as cell, tissue or organ donors. In this reviewer's opinion the specific research is of high importance since almost all research works depend on volunteers. The only comment this reviewer has is that the authors could have also asked the participants what would help them be more positive about participating in volunteering for research purposes. Since such a question was omitted, it would be interesting if the authors could share in their conclusions their opinion regarding how we could "persuade" both healthy donors and patients to participate in research projects. I take the liberty to share that describing the potential benefits of a project has helped me and my team to recruit donors in the past.

The english language is fine. There are only minor mistakes.

Author Response

Dear Reviewer,

First of all, we would like to express our gratitude to you for giving us the opportunity to revise and resubmit our paper. We are also indebted to your valuable suggestions and helpful comments. We hope that this revised paper is more consistent owing to your willingness to help. We have been convinced by all of your arguments and we are very grateful to you for pointing these things out. We believe that we have also answered all of your questions. Still, should it happen that we have missed and/or misunderstood any vital comment, we would be more than happy to promptly address any edits and further revise our article.

Below, we detail the changes that we have made in accordance with your suggestions and comments (please note that in the revised manuscript we have taken the liberty of marking the most important changes in the colour blue to facilitate their checking).

  1. Reviewer’s comment:

The authors conducted an interesting study regarding the will of Polish healthcare students to participate in research activities as cell, tissue or organ donors. In this reviewer's opinion the specific research is of high importance since almost all research works depend on volunteers. The only comment this reviewer has is that the authors could have also asked the participants what would help them be more positive about participating in volunteering for research purposes. Since such a question was omitted, it would be interesting if the authors could share in their conclusions their opinion regarding how we could “persuade” both healthy donors and patients to participate in research projects. I take the liberty to share that describing the potential benefits of a project has helped me and my team to recruit donors in the past.

Authors’ response:

Thank you, we appreciate your positive feedback. At the same time, while we greatly appreciate your suggestion we wish to explain that since in our research we have assessed how medical and healthcare students’ willingness to donate interacts with certain type of tissues and body parts, and/or biomedical research, we believe that providing suggestions on how to persuade “healthy donors and patients” goes beyond the scope of this work and could be rather a subject for a different article. However, following you apt suggestion in the Conclusion section we have provided some suggestions that could enhance future healthcare professionals’ awareness and interest in biomedical research and donation for research purposes. This has been done on page 13, lines 452-468. Thus, after revision it now says (the revised part is marked in the colour blue):

“Thus, apart from public education campaigns that address the role and function of biomedical research there is still an urgent need to raise medical and healthcare students’ awareness about their important role in research activities. This is of key importance, because as future healthcare workers they will be responsible not only for caring over patients but also for the promotion of biomedical research. Put together, to overcome medical and healthcare students’ ambivalent attitudes toward tissues donation for research purposes we recommend that they should receive more education and awareness about the importance of furthering biomedical research and precision medicine. To achieve that the following guidelines should be implemented:

  1. In order to rise medical and healthcare students’ awareness and interest in biomedical research and donation, university curricula in Poland should include teaching programs on such issues.
  2. Because biomedical research and donation for research purposes provoke many ethical and moral dilemmas among the donors, future healthcare professionals should be also familiarized with ethical and legal framework for donation, (non-verbal) communication and active listening skills, the role of cultural belief systems on tissue and organ donation, and education of general public on donation.
  3. Because many healthcare professionals feel reluctant to approach their patients who could possibly share their HBM for research purposes, or their families, medical and healthcare students should be trained how to effectively recruit tissue and organ donors.
  4. Additionally, e-learning modules, counting tutorial lecturer videos, webinars, online postgraduate specialization courses, healthcare professional awareness and training programs on biomedical research and donation should be also organized.“

We hope this explanation satisfies your concerns. At the same time, we are grateful to you for bringing our attention to this point.

All in all, again we wish to thank you for all your valuable suggestions and willingness to help us in improving our manuscript. We hope that our revisions satisfy your concerns and that this revised paper is more consistent owing to your willingness to help. However, if we have missed and/or misunderstood any vital comment, we would be more than happy to further revise our article.

Sincerely,

Authors

Round 2

Reviewer 1 Report

Thank you for submitting the new version of the manuscript. The revisions made by the authors are indeed substantial, focusing more on explanations rather than introducing new analysis methods and results.

As I mentioned in the previous review report, the dominant factors influencing willingness to donate are religion, gender, and various social parameters. If you would like to explore correlations or clusters related to these dominant factors, it's essential to account for their effects. Using university-level analysis methods such as the chi-square test, cluster analysis, and correlation analysis alone may not be sufficient to substantiate the results, and their reliability might be in question. Multivariate analysis, cross-validation between different methods, and robustness/confidence analysis are necessary steps to ensure the validity of the findings.

I'm afraid I cannot recommend accepting this manuscript in its current version unless the authors make some significant improvements.

Author Response

Dear Reviewer,

First of all, we would like to thank you for giving us the opportunity to further revise and resubmit our paper. Below, we detail the your comments and our responses. At the same time, we wish to stress that while in this revised manuscript we have taken the liberty of marking additional changes in colour purple to facilitate their checking,  changes made during the first round of revision are still marked in the following colours: red for the Reviewer#1; green for the Reviewer #2 , and blue for the Reviewer #3).

  1. Reviewer’s comment:

As I mentioned in the previous review report, the dominant factors influencing willingness to donate are religion, gender, and various social parameters.

Authors’ response:

While we greatly appreciate your comment, again we wish to stress that the purpose of our study was not to assess the dominant sociodemographic factors that influence peoples’ willingness to donate their biospecimens for research purposes, since such analysis has already been conducted by many other researchers, including some in Poland. Instead, our research aimed at investigating how future healthcare professionals bound certain types of tissues and biomedical research and how such a cluster thinking can affect their readiness to participate in the collection and management of biospecimens. At the same, since we are aware that that there are some important socio-demographic factors, including those you have mentioned in your comment, that influence people’s willingness to donate for the purposes, in the very first paragraph of the Discussion section (lines 278-286) we such attitudes of research is driven by their knowledge about the donation system and biomedical research, declared system of values, religious beliefs, and demographic characteristics, including age, sex, education, and family and ethnic background, but also previous experiences with the healthcare system, social trust in physicians and scientists, and perceived benefits and risks associated with biomedical research.

However, as already mentioned in our previous response, since most research on donation for research purposes focus on people’s knowledge about biomedical research, willingness and motivations to share biospecimens for research purposes, preferred model of consent or trust towards various research institutions, but do not ask the questions that were crucial for our research (i.e. the type of tissues or body parts to be donated and the type of biomedical research to be conducted) we do not refer to those studies even though they do describe socio-demographic factors that affect people’s general readiness to donate. Thus, taking under consideration the aim of our study and three questions asked, still we do not see the point in describing the correlation between “religion, gender, and various social parameters” and such issues as people’s knowledge on donation, willingness and motivations to share biospecimens for research purposes, preferred model of consent, ownership and profit or trust towards biomedical institutions, since it goes beyond the scope of this work and could be rather a subject for a different article.

However, following your recommendation to previously added paragraphs describing  the correlation between sociodemographic factors on people’s willingness to donate various types of tissues or body organs (pages 11-12, lines 385-407), also in this revised manuscript we have added yet another paragraph describing factors affecting peoples’ attitudes towards donation of brain and eyes. This has been done on page 12 (lines 408-416). Thus, after revision it now says:

“In parallel, this study demonstrates that factors such as gender, faculty, and academic year exerted discernible impacts on students' proclivity towards tissue donation. Notably, statistically significant distinctions were detected concerning the specific tissue types and research domains for which students were inclined to offer their donations. Across the majority of research categories and various tissue types, male students and those pursuing medical and biotechnology disciplines exhibited the highest levels of willingness to contribute, whereas their female counterparts and students specializing in physiotherapy displayed comparatively lower inclinations toward such contributions

However, also Goodson and Vernon demonstrated that people’s willingness to donate is influenced by their sex, as females were more willing to donate most type of body tissues for clinical research (including eye, head and neck tissue, heart, brain, lung or bones), but men were less restricted about the type of research their body tissues or organs would be used for (i.e. general knowledge, genetic disorders, testing medicines) [3]. Similarly, Lewis et al. [4] showed that the willingness to donate spare eggs and spare embryos was much more common among men who were under 55 years, from a higher socioeconomic group, white, not at all or moderately religious, and had tissue removed during a medical procedure. On the other hand, women who supported donation of reproductive tissues were be from a higher socioeconomic group, white and had no religious affiliation. While preferences in Polish population did not differ statistically for sex, age, education, religiousness or trust in other people, it was engagement in charity, financial situation and trust in physicians and scientists that influenced people’s willingness to donate [15]. Other research suggest that also education level associated with higher awareness of bioethics affects people willingness to donate excess surgical tissue [23].

Boise et al. found out that among diverse ethnic groups older age and Latino ethnicity were positive predictors for the willingness to become a brain donation, while African/African American race was a negative predictor [47]. Another study by Hussen et al. demonstrated that the willingness to donate eyes among Ethiopians was associated with religion, educational level, and awareness [49]. (High) school education was also found as a positive predictor for eyes donation in India, while religious reasons and cultural beliefs were the main reasons for lack of willingness to eyes donation [50]. Finally, Chinese older women living in Chicago’s Chinatown were more eager to donate hair or nail than blood [52].”

We hope this additional paragraph satisfies your concerns.

  1. Reviewer’s comment:

If you would like to explore correlations or clusters related to these dominant factors, it's essential to account for their effects. Using university-level analysis methods such as the chi-square test, cluster analysis, and correlation analysis alone may not be sufficient to substantiate the results, and their reliability might be in question. Multivariate analysis, cross-validation between different methods, and robustness/confidence analysis are necessary steps to ensure the validity of the findings.

Authors’ response:

While we appreciate your additional comment, we are somehow perplexed by your accusation that our work lacks multivariate analysis. In fact, already in the first version of our paper we had incorporated Principal Component Analysis (with a Promax rotation variant), which is a part of multivariate analysis. Moreover, following your recommendation, upon further development of our techniques, we introduced another multivariate analysis method - Cluster Analysis (using the kMeans variant).

            Your accusation of a lack of cross-validation is also surprising. Setting aside the fact that many of our findings had p-values < 0.001, making them highly unlikely to be mere chance, there is an additional issue with this claim. Our findings were, in any case, cross-validated by processing the same data through several techniques, including correlation, PCA, cluster analysis and chi-squared test. That was our key finding, while socio-demographic predictors of belonging to a particular group were an additional finding.

It is also surprising that you characterize our techniques as ‘university-level analysis methods’ without specifying which particular techniques we should apply in this case. Especially, that, as noted above, even though in the original version of our manuscript we had already applied two multivariate analyses and three methods allowing for cross-validation of results, during the first round of revision, following the Reviewers suggestion we have incorporated cluster analysis as yet another technique. Thus, put together we have used four different techniques: Principal Component Analysis, Cluster Analysis, correlation and chi-squared test.

What makes your claim even more surprising is while none of the reviewers raised objections about the literature we cited in our research, to our best knowledge none of the studies on the topic used so many different statistical techniques we did in our research. On the contrary, most studies relied on one or two techniques, including those we used in our study: chi-squared test and PCA. To justify our argument below we list 21 previous works on the topic we cited in our paper. Among them, 6 did not employ a single statistical test; however, in all but one case, this omission was justified as they were qualitative works. Eleven works used a single test, and the most commonly used test was the chi-squared test. (Here, we are counting the use of the chi-squared test for binary data and the t-test for Likert scale questions as one, although counting them separately would also increase the number of tests that we have applied.) Only four works used two tests, with the most popular combination being the chi-squared test and logistic regression. In the analysed works, none used three types of tests, while our work stands out by applying four types of tests: Principal Component Analysis, Cluster Analysis, correlation and chi-squared test, making it an outlier. Additionally, when assessing whether someone’s techniques were “university-level analysis methods”, we are the only group that used Python, which is right now the default tool for more sophisticated analysis in data science and machine learning. The list:

  1. Goodson, M.L.; Vernon, B.G. A study of public opinion on the use of tissue samples from living subjects for clinical research.

-standard error of percentage

  1. Lewis, C.; Clotworthy, M.; Hilton, S.; Magee, C.; Robertson, M.J.; Stubbins, L.J.; Corfield, J. Public views on the donation and use of human biological samples in biomedical research: a mixed methods study.

-chi-squared

  1. Pawlikowski, J.; Wiechetek, M.; Majchrowska, A. Associations between the willingness to donate samples to biobanks and selected psychological variables.

-correlation

-linear regression – though as they show (R2 = 0.034), that their model explains 3.4% of observed variability, they are kind of showing that this technique did not yield especially useful results

  1. Majchrowska, A.; Wiechetek, M.; Domaradzki, J.; Pawlikowski, J. Social differentiation of the perception and human tissues donation for research purposes.

-ANOVA

-PCA

  1. Kintossou, A.K.; N’dri, M.K.; Money, M.; Cissé, S.; Doumbia, S.; Soumahoro, M.K.; Coulibaly, A.F.; Djman, J.A.; Dosso, M. Study of laboratory staff’ knowledge of biobanking in Côte d’Ivoire.

-chi-squared

  1. Abdelhafiz, A.S.; Sultan, E,A.; Ziady, H.H.; Sayed, D,M.; Khairy, W.A. Knowledge, perceptions and attitude of Egyptian physicians towards biobanking issues.

-chi-squared

  1. Tozzo, P.; Fassina, A.; Caenazzo, L. Young people’s awareness on biobanking and DNA profiling: results of a questionnaire administered to Italian university students.

-chi-squared

  1. Merdad. L.; Aldakhil, L.; Gadi, R.; Assidi, M.; Saddick, S,Y.; Abuzenadah, A.; Vaught, J.; Buhmeida, A.; Al-Qahtani, M.H. Assessment of knowledge about biobanking among healthcare students and their willingness to donate biospecimens.

-chi-squared / t-test

  1. Khatib, F.; Jibrin, D.; Al-Majali, J.; Elhussieni, M.; Almasaid, S.; Ahram, M. Views of university students in Jordan towards biobanking.

-chi-squared

  1. Igbe, M.A.; Adebamowo, C.A. Qualitative study of knowledge and attitudes to biobanking among lay persons in Nigeria

-none (focus group)

  1. Heredia, N,I.; Krasny, S.; Strong, L.L.; Von Hatten, L.; Nguyen, L.; Reininger, B.M.; McNeill, L.H.; Fernández, M.E. Community perceptions of biobanking participation: A qualitative study among Mexican-Americans in three Texas cities. Public Health Genom. 

-none (focus group)

  1. Goddard, K.A.; Smith, K.S.; Chen, C.; McMullen, C.; Johnson, C. Biobank recruitment: Motivations for nonparticipation

-chi-squared

-logistic regression? (claim to have used it, does not clearly show any results)

  1. Sanderson, S.C.; Brothers, K.B.; Mercaldo, N.D.; Clayton, E.W.; Matheny, A.A.H.; Aufox, S.A.; Brilliant, M.H.; Campos, D.; Carrell, D.S.; Connolly, J.; Conway, P.; Fullerton, S.M.; Garrison, N.A.; Horowitz, C.R.; Jarvik, G.P.; Kaufman, D.; Kitchner, T.E.; Li, R.; Ludman, E.J.; McCarty, C.A.; McCormick, J.B.; McManus, V.,D.; Myers M.,F.; Scrol, A.; Williams, J.L.; Shrubsole, M.J.; Schildcrout, J.S.; Smith, M.E.; Holm, I.A. Public attitudes toward consent and data sharing in biobank research: A large multi-site experimental survey in the US.

-chi-squared

-logistic regression

  1. Lipworth, W.; Morrell, B.; Irvine, R.; Kerridge, I. An empirical reappraisal of public trust in biobanking research: Rethinking restrictive consent requirements. 

-none (semi-structured survey)

  1. Schwartz, M.D.; Rothenberg, K.; Joseph, L.; Benkendorf, J.; Lerman, C. Consent to the use of stored DNA for genetics research: a survey of attitudes in the Jewish population

-chi-squared/t-test 

- ANOVA

  1. Lemke, A.A.; Wolf, W.A.; Hebert-Beirne, J.; Smith, M.E.; Public and biobank participant attitudes toward genetic research participation and data sharing

-none (focus group)

  1. Barnes, R.; Votova, K.; Rahimzadeh, V.; Osman, N.; Penn, A.M.; Zawati, M.H.; Knoppers, B.M. Biobanking for genomic and personalized health research: Participant perceptions and preferences

-none, just presenting numerical findings

  1. Boise, L.; Hinton, L.; Rosen, H.J.; Ruhl, M.C.; Dodge, H.; Mattek, N.; Albert, M.; Denny, A.; Grill, J.D.; Hughes, T.; Lingler, J.H.; Morhardt, D.; Parfitt, F.; Peterson-Hazan, S.; Pop, V.; Rose, T.; Shah, R.C. Willingness to be a brain donor: A survey of research volunteers from 4 racial/ethnic groups

-chi-squared / t-test

-logistic regression

49 Hussen, M.S.; Gebreselassie, K.L.; Woredekal, A.T.; Adimassu, N.F. Willingness to donate eyes and its associated factors among adults in Gondar town, North West Ethiopia.

-Binary logistic regressions

  1. Marmamula, S.; Priya, R.; Varada, R.; Keeffe, J.E. Awareness on Eye Donation in the North-eastern State of Tripura, India - The Tripura Eye Survey.

-logistic regression

  1. Simon, M.A.; Tom, L.S.; Dong, X. Knowledge and beliefs about biospecimen research among Chinese older women in Chicago’s Chinatown.

-none (focus group)

All in all, again we wish to stress that since this paper aimed at investigating the correlation between medical and healthcare students’ willingness to donate a biospecimen and the type of tissues to be donated and the type of biomedical research to be conducted, we have focused on showing how students bound certain types of tissues and biomedical research and how it affects their willingness to donate. For that reason in the Discussion section we have focussed only on those research that asked questions that were crucial for our research (i.e. the type of tissues or body parts to be donated and the type of biomedical research to be conducted).

More importantly, against your claim, since in our analysis we have used four different techniques: Principal Component Analysis, Cluster Analysis (which was requested by the Reviewer), correlation and chi-squared test it is hard for us to understand your critique of methods used.

We do hope that our explanations clarify our stance.

Yours sincerely,

Authors

Reviewer 2 Report

The authors have significantly made the changes requested

Author Response

Dear Reviewer,

First of all, we would like to express our gratitude to you for giving us another opportunity to further revise and resubmit our paper. We are also indebted to your valuable suggestion regarding sampling method which was improved in this version of our manuscript. We hope that this revised paper is more consistent owing to your willingness to help. Still, should it happen that we have missed and/or misunderstood any vital comment, we would be more than happy to promptly address any edits and further revise our article.

At the same time, we wish to stress that while in this revised manuscript we have taken the liberty of marking additional changes in colour purple to facilitate their checking,  changes made during the first round of revision are still marked in the following colours: red for the Reviewer#1; green for the Reviewer #2 , and blue for the Reviewer #3).

  1. Reviewer’s comment:

The authors have addressed majority of the point raised. However, the sampling size was not properly addressed.

Abstract The sampling method was not still stated 

Authors’ response:

We are grateful to you for this remark because it has helped us in clarifying this important methodological issue. Thus, we wish to explain that since, this study was designed as a population survey there was no sampling in this study as all students enrolled in PUMS received an invitation to participate in the study. At the same time, following your recommendation in the abstract we have added this information. This has been done on page 1, lines 21-24. Thus after revision it now says:

“A population survey among medical and healthcare students enrolled at the Poznan University of Medical Sciences was conducted on their attitudes toward the donation of human biological material for research purposes. A total of 1,500 students responded and completed the survey”

Additionally, in the Data Collection section we have added information about the sample size. Thus, we explain that although of all students approached only 1,500 (25.72%) responded and completed the survey, social or demographic features of our sample were similar to those in general population of students enrolled at PUMS. This has been done on page 3, lines 125-130. Thus after revision it now says:

“2.4. Data Collection

The structured questionnaire was posted in an online platform and distributed to all students of all faculties at PUMS (medicine, medical sciences, pharmacy, and health sciences) via a communication platform used at PUMS for educational purposes during the COVID-19 pandemic. Since this study was designed as a population survey all students enrolled in PUMS received an invitation email and were informed about the study’s purpose, as well as the voluntary, anonymous, and confidential character of the study. Out of all the 5,830 students approached, a total of 1,500 (25.72%) students responded and completed the survey. At the same time, social or demographic features of our sample were similar to those in general population of students enrolled at PUMS.

All participants completed self-administered, computer-assisted questionnaires using electronic devices. Questionnaires took approximately 20 minutes to complete and were collected anonymously.”

Finally, since we are aware that the number of respondents was rather small and the results cannon be generalized to the entire population of medical and healthcare students in Poznan in the Limitation section (page 13, lines 419-422) we have provided as the major limitation of our survey.

We hope this explanation satisfies your concerns. At the same time, we are grateful to you for bringing our attention to this point.

All in all, again we wish to thank you for all your valuable suggestions and willingness to help us in improving our manuscript. We hope that our revisions satisfy your concerns and that this revised paper is more consistent owing to your willingness to help. However, if we have missed and/or misunderstood any vital comment, we would be more than happy to further revise our article.

Sincerely,

Authors
